# Study on the Metabolic Transformation Rule of Enrofloxacin Combined with Tilmicosin in Laying Hens

**DOI:** 10.3390/metabo13040528

**Published:** 2023-04-06

**Authors:** Jingchao Guo, Liyun Zhang, Yongxia Zhao, Awais Ihsan, Xu Wang, Yanfei Tao

**Affiliations:** 1National Reference Laboratory of Veterinary Drug Residues (HZAU), MAO Key Laboratory for Detection of Veterinary Drug Residues, Huazhong Agricultural University, Wuhan 430070, China; 2MAO Laboratory for Risk Assessment of Quality and Safety of Livestock and Poultry Products, Huazhong Agricultural University, Wuhan 430070, China; 3Department of Biosciences, COMSATS University Islamabad, Sahiwal Campus, Islamabad 45550, Pakistan

**Keywords:** enrofloxacin, tilmicosin, laying hen, metabolic transformation rule

## Abstract

There is often abuse of drugs in livestock and poultry production, and the improper use of drugs leads to the existence of a low level of residues in eggs, which is a potential threat to human safety. Enrofloxacin (EF) and tilmicosin (TIM) are regularly combined for the prevention and treatment of poultry diseases. The current studies on EF or TIM mainly focus on a single drug, and the effects of the combined application of these two antibiotics on EF metabolism in laying hens are rarely reported. In this study, liquid chromatography-tandem mass spectrometry (LC-MS/MS) was used to determine the residual EF and TIM in laying hens and to investigate the effect of TIM on the EF metabolism in laying hens. In this paper, we first establish a method that can detect EF and TIM simultaneously. Secondly, the results showed that the highest concentration of EF in the egg samples was 974.92 ± 441.71 μg/kg on the 5th day of treatment. The highest concentration of EF in the egg samples of the combined administration group was 1256.41 ± 226.10 μg/kg on the 5th day of administration. The results showed that when EF and TIM were used in combination, the residue of EF in the eggs was increased, the elimination rate of EF was decreased, and the half-life of EF was increased. Therefore, the use of EF and TIM in combination should be treated with greater care and supervision should be strengthened to avoid risks to human health.

## 1. Introduction

China is one of the largest countries of egg production. According to a few reports, the egg production in China has reached more than 20 million tons and China’s per capita egg consumption has been growing steadily [1,2]. In this case, while meeting people’s food needs, eggs inevitably have certain safety problems [3]. In the livestock and poultry industry, veterinary drugs such as antibiotics and antiparasitics are often used for disease prevention and treatment [4,5]. However, because of the intensive breeding of laying hens, the excessive addition of antibiotics, the failure to maintain the prescribed antibiotic withdrawal period before selling eggs and other foods, etc., the antibiotic residues in laying hens may exceed the standard, leading to the increased risk of egg food safety problems [6]. Despite regulations in the European Union, the United States, and China that prohibit the use of antibiotics in eggs, antibiotic residues are still detected in eggs [7]. Because the distribution of veterinary drug residues does not always follow established patterns, these compounds in egg derivatives must be monitored to determine the safety of the final product.

Enrofloxacin (EF), one of the members of the 3rd-generation synthetic antibiotics for fluoroquinolones (FQs), was the first antibiotic from the fluoroquinolone group for the use in animals [8]. As a new synthetic antimicrobial agent, EF has a wide antibacterial spectrum, with strong antibacterial activity [9]. In addition, EF has few toxic side effects, so it has been widely used in the production process for a long time [10]. Due to the lack of knowledge of some farmers, the long-term use also results in the residue of EF in animal products, which not only brings great harm to human health but also affects the development of animal husbandry. Moreover, the residual issues have attracted the attention of the livestock industry [11]. According to the study reports, the most commonly isolated residue from antibacterial substances was EF in egg samples [5]. Therefore, for a long time in the past, the problem of EF residue caused wide concern in the food and feed industry. Although few antibiotics are approved for use in the laying industry, the detection of EF residues in eggs is a significant concern [12]. For example, the presence of EF residues in eggs has been widely reported in commercial samples [13,14]. In addition, in poultry systems, FQ can be detected due to the handling or off-label and unnecessary use of animals at subtherapeutic levels. Considering that egg consumption is a prevalent practice throughout the world, it is crucial to analyze the presence of EF and its residues in egg samples and derivatives.

Tilmicosin (TIM) is a macrolide antibiotic for livestock and poultry synthesized from tylomycin [15]. Currently, it is mainly used to treat common bacterial infections and diseases caused by mycoplasma infection in livestock and poultry breeding [16]. TIM has broad-spectrum antibacterial properties and has a strong inhibitory effect on most *Gram bacteria* including negative and positive bacteria, as well as a strong antibacterial ability and activity [16]. *Mycoplasma gallinarum* is extremely sensitive to TIM, which has strong permeability in lung tissue, so TIM is one of the ideal drugs for the clinical control of respiratory tract infections [17]. Based on face-to-face surveys of veterinary antibiotic use on chicken farms, TIM was occasionally administered to laying hens at high doses for short periods of time, especially on small farms [18]. In 2014–2015, Takahiro et al. monitored the residual levels of antibiotics in Vietnamese eggs, and it was found that TIM was detected in 3 samples out of 16 samples, suggesting that TIM was commonly used in laying hens in Vietnam [19].

Because of the complexity of the infection and problems such as drug resistance, a single drug cannot provide effective and safe treatment [20]. Studies have reported that combined antibacterial drugs have a better therapeutic effect on specific multiple infections [21,22]. However, the composite approach also presents challenges. The combination of antibiotics can interact with each other in pharmacodynamics, pharmacokinetics, physicochemical properties, etc. An improper combination of antibiotics can reduce the efficacy and even cause the poisoning or death of poultry. In the practical application of aquaculture, EF and TIM are often used in combination to avoid drug resistance caused by a single drug and improve the therapeutic effect [23]. EF and TIM have long been used in combination to treat chronic respiratory infections in chickens [24]; drug combinations are bound to produce drug interactions [25]. However, little has been reported about the effects of the residues of the two drugs’ combination in eggs. In view of the increasing consumption of eggs worldwide, it is of great significance to analyze the use of drug combinations in eggs.

The aim of this study was to establish a new liquid chromatic-tandem mass spectrometry (LC-MS/MS) method to investigate the changes in the EF and TIM residues in eggs during the combination of EF and TIM and to evaluate the feasibility of the combination of EF and TIM in poultry and whether it poses a potential threat to human health. In production practice, a drug combination will inevitably cause a drug interaction, which may lead to toxic side effects. Therefore, it is of great significance to conduct extensive and in-depth studies on the application of EF and TIM in laying hens to provide a theoretical basis for the combined application of EF and TIM while also helping to avoid potential risks to human health.

## 2. Materials and Methods

### 2.1. Chemicals

EF standard (98%) and TIM (95.3%) were obtained from China Institute of Veterinary Drugs Control. Methanol and acetonitrile (chromatography pure) were obtained from Sinopharm Chemical Reagent Co., LTD. (Shanghai, China). Formic acid (chromatograph pure) was obtained from MACKUN Corporation (Batch number: F809712).

### 2.2. Standard Solutions

The 0.1% formic acid water: Take 1 mL (99.9%) formic acid and dilute it with ultra-pure water in a 1000 mL bottle. Before use, degas it in ultrasonic cleaning machine for 20 min. The 1% formic acid acetonitrile: Remove 10 mL formic acid (99.9%) to acetonitrile reagent and dilute in 1000 mL bottle. Before use, degassing was carried out by ultrasound for 20 min. TIM Standard reserve solution: 10.49 mg TIM standard (95.3%) was accurately weighed in 10 mL brown volumetric bottle, dissolved with methanol and fixed volume. Sealed and stored at 4 °C away from light. TIM standard working solution: The TIM standard reserve solution was accurately measured in a brown volumetric bottle with a pipet, then diluted and volumetric with a mobile phase, and prepared into a 10 µg/mL standard working solution, sealed and stored at 4 °C away from light.

### 2.3. Animals

In this experiment, fifty healthy and high-yield white laying hens (35 weeks of age) with body weight of 1.7 kg were selected. The hens were kept in exclusive animal houses where the temperature is kept at 25 °C. Before the experiment began, all the animals were allowed a seven-day acclimation period, given an antimicrobial free diet and free water. Monitored the physiological condition of the hens daily.

### 2.4. Animal Experimental Design

#### 2.4.1. Animal Grouping and Sampling

A total of 50 healthy laying hens with average weight of 1.7 kg were randomly divided into 4 groups: blank control group, EF single administration group, TIM single administration group, EF combined with TIM administration group. The physiological condition of laying hens was monitored daily, and no clinical symptoms were observed after 1 week of acclimation. Egg samples were collected and tested as blank samples without drugs.

In TIM group: 10% TIM soluble powder was diluted with drinking water (750 mg/L) and administered with drinking water for 3 consecutive days. The blank control group was given the same amount of drinking water. Collected eggs daily (no less than 10) from the first dose, and at least 10 eggs were collected daily at 4 h (zero withdrawal period), 3, 5, 8, 10, 12, and 15 days after the last dosing. Egg samples were stored at −80 °C. The egg samples during the administration period and 4 h (zero withdrawal period), 3, 5, 8, 10, 12, and 15 days after the last administration were analyzed and detected.

In EF group: EF soluble powder was diluted with drinking water (750 mg/L) and administered with drinking water for 5 consecutive days. The blank control group was given the same amount of drinking water. Collected eggs daily (no less than 10) from the first dose, and at least 10 eggs were collected daily at 4 h (zero withdrawal period), 3, 5, 8, 10, 12, and 15 days after the last dosing. Egg samples were stored at −80 °C. The egg samples during the administration period and 4 h (zero withdrawal period), 3, 5, 8, 10, 12, and 15 days after the last administration were analyzed and detected.

In EF combined with TIM group: 10% TIM soluble powder and EF soluble powder were diluted with drinking water (750 mg/L) and administered with drinking water for 5 consecutive days. The blank control group was given the same amount of drinking water. Collected eggs daily (no less than 10) from the first dose, and at least 10 eggs were collected daily at 4 h (zero withdrawal period), 3, 5, 8, 10, 12, and 15 days after the last dosing. Egg samples were stored at −80 °C. The egg samples during the administration period and 4 h (zero withdrawal period), 3, 5, 8, 10, 12, and 15 days after the last administration were analyzed and detected.

#### 2.4.2. Sample Pretreatment

Each dosing group collected at least 10 eggs per day. The homogenized egg samples were divided into clean EP tubes and marked and stored below −80 °C for future use. The 1.00 g sample was accurately weighed, 5 mL 1% acetonitrile was precisely added, and the sample was violently oscillated in the oscillator for 10 min. Ultrasonic extraction was performed at 2–8 °C for 15 min. Supernatant was obtained by centrifugation at 10,000 r/min at 4 °C for 10 min. A 1 mL 80% acetonitrile cleaning column. The supernatant of the sample was passed through the column at a certain velocity. The filtrate was collected and dried with nitrogen. The initial mobile phase was determined by LC-MS/MS.

#### 2.4.3. Method for Detection of EF in Combination with TIM in Eggs

The chromatographic column, mobile phase, and gradient elution of mobile phase used in this study are the same as those reported by Zhang et al. [26]. The mass numbers of determined quantitative ions and auxiliary qualitative ions are shown in Table 1.

### 2.5. Method Validation

In this study, method specificity, the limit of detection [27], the limit of quantification (LOQ) [28], matrix and standard curve matching [29], accuracy, and precision are shown in the Appendix A.

### 2.6. Samples Testing

The collected egg samples were processed according to the method in Section 2.4.2, and then after the 0.22 µm filtration membrane was passed, it was instrumentally tested according to the conditions in Section 2.4.3. The corresponding peak areas of EF, CIP, and TIM were recorded, and the concentrations of EF, CIP, and TIM in eggs were calculated using the standard curve regression equation. For samples beyond the linear range, it is necessary to test after dilution.

### 2.7. Data Analysis

The concentrations of EF and TIM were quantified by matrix-matching calibration curves. Calculate ddescriptive statistical parameters such as mean, standard deviation (SD), and coefficient of variation (CV). The residual consumption curves of EF and TIM in eggs were estimated by linear regression. The half-lives (t_1/2_) of EF and TIM were calculated graphically in the elimination phase by fitting linear regression. The differences between the TIM and EF-TIM were analyzed by one-way ANOVA. Statistical significance was set at *p* < 0.05, and *p* < 0.01 was considered to be strongly significant.

Determination of sample concentration: The sample concentration is calculated by external standard method, and the response values of standard solution and sample solution are within the range of instrument detection. The unit concentrations of EF, CIP, and TIM in eggs are calculated by the following formula:$$X=\frac{AC_{S}V_{1}V_{3}D}{A_{S}V_{2}M}\;$$

In the formula:X-EF/CIP/TIM concentration per unit weight (volume) of a sample, expressed in μg/kg;*A*: peak area of the loading solution.*A_S_*: peak area of the standard control solution.*C_S_*: the concentration of the standard reference solution, expressed in μg/kg.*V_1_*: total volume of extract, expressed in mL.*V_2_*: volume of extract removed for nitrogen blowing concentration, expressed in mL.*V_3_*: constant volume after concentration, expressed in mL.*M*: mass of sample, expressed in g.*D*: dilution ratio before determination.

## 3. Results and Analysis

### 3.1. Assessment of Methodology

#### 3.1.1. Specificity

The blank egg homogenate samples were added with a 10 μg/kg (limited quantitation concentration) mixed standard solution of EF, CIP, and TIM, respectively. The samples were processed and purified as shown in Section 2.4.2 and detected by LC-MS/MS. The results of the three drugs are shown in Figure 1. The chromatographic results show that there are no stray peaks or interference peaks near the drugs to be detected. The specificity of this method is good and meets the assessment requirements of the quantitative methodology.

#### 3.1.2. Minimum Detection Limit and Minimum Quantification Limit

The standard mixed solutions of EF, CIP, and TIM were added to the treated egg homogenate samples. The result of the measurement is determined as the lowest detection limit according to the 3-times signal-to-noise ratio: the detection limit of EF, CIP, and TIM in the egg homogenate samples was 5.0 μg/kg; the result of the measurement is determined as the minimum limit of quantification according to the 10-times signal-to-noise ratio: the minimum limit of quantification (LOQ) of EF, CIP, and TIM in the egg homogenate samples was 10.0 μg/kg for all, and the accuracy and precision were greater. The LOD and LOQ in the egg samples established in this study were 5.0 µg/kg and 10.0 µg/kg, respectively.

#### 3.1.3. Matrix Standard Curve

The matrix-matching concentration of the EF, CIP, and TIM mixed standard working solution was determined by the LC-MS/MS method under the established mass spectrum conditions. Weighed a 1.00 ± 0.01 g homogenized blank sample in a 10 mL centrifuge tube, diluted the mixed standard working liquid with blank extract, and then passed a 0.22 µm filtration membrane for machine detection. The matrix-matching concentration of the egg samples was in the range of 10–500 μg/kg, with the tissue concentration represented on the horizontal axis and the peak area of the ions represented on the vertical axis. There is a good linear relationship between the drug concentration and the ion peak area. Table 2 shows the linear regression equations and correlation coefficients of the standard curves for the three drugs.

#### 3.1.4. Accuracy and Precision

The precision and accuracy were expressed by the relative standard deviation and recovery rate, respectively. The appropriate concentration of the standard solution was added into the blank sample, and the concentrations were low, medium, and high (10, 50, and 100 μg/kg), respectively. The mixed standard solution of EF, CIP, and TIM was added to the poultry egg samples to prepare the concentration levels of 10, 50, and 100 μg/kg. According to the sample treatment method, the extraction, purification, and then the HPLC/MS tandem detection analysis were performed. There were five replicates per concentration and three separate tests were performed. The results are shown in Table 3: at the three concentration levels, the average in-batch recoveries were 77.78–95.51%, and the average inter-batch recoveries were 80.12–92.33%. The in-batch coefficients of the variation in EF, CIP, and TIM were 1.63–9.10%, and the coefficient of variation between the batches of EF, CIP, and TIM ranged from 1.04% to 7.28%. The method established in this experiment meets the requirements of the methodology assessment in the detection of residues in egg samples and can be used for the analysis of EF, CIP, and TIM residues in eggs.

### 3.2. Residue Elimination Features of EF in Eggs

The residual concentrations of EF in the egg samples at different time points are shown in Table 4, Table 5, Table 6 and Table 7. The EF single administration group: The peak concentrations of EF and CIP in the egg samples were 974.92 ± 441.71 μg/kg and 121.51 ± 14.38 μg/kg on the 5th day of administration (Table 4). The residual concentrations of EF and CIP decreased to 577.12 ± 101.85 μg/kg and 42.12 ± 15.42 μg/kg, respectively, on day 3 after withdrawal (Table 5). The EF content was 121.54 ± 19.12 μg/kg and 68.64 ± 8.68 μg/kg on day 8 and day 10 after drug withdrawal, respectively, and CIP had been eliminated on day 8. On the 12th day, the EF concentration decreased to 45.78 ± 8.65 μg/kg. On the 15th day, the EF residue was eliminated, and the t_1/2_ of EF and CIP were 1.13 d and 1.39 d, respectively (Table 8).

The EF combined administration group: The peak concentrations of EF and CIP in the egg samples were 1256.41 ± 226.10 μg/kg and 93.22 ± 22.79 μg/kg on the 5th day of administration (Table 6). The concentrations of EF and CIP decreased to 715.55 ± 190.01 μg/kg and 42.58 ± 7.43 μg/kg, respectively, on day 3 after withdrawal (Table 7). The EF content was 153.29 ± 25.48 μg/kg on day 8 and 89.51 ± 13.43 μg/kg on day 10 after drug withdrawal, and CIP had been eliminated on day 8. On the 12th day, the EF concentration decreased to 58.71 ± 7.41 μg/kg. On the 15th day, the EF residue was eliminated, and the t_1/2_ of EF and CIP were 1.17 d and 1.14 d, respectively. On the whole, the distribution concentration of EF was higher in the combined group than in the single group, and the elimination rate of EF in the combined group was slower than that in the single group (Figure 2 and Figure 3).

With time as the horizontal axis and the EF content in the egg samples as the vertical axis, the fitting was carried out. The drug concentration at the next three time points where the drug could be detected was logarithmic and then the regression analysis was made with time to obtain the elimination parameters of EF in the three samples. If the concentration is detected at less than three time points, the elimination parameter cannot be calculated. The elimination parameters of EF in the whole egg and the drug–timing curve are shown are shown in Table 8.

### 3.3. Residue Elimination Characteristics of TIM in Eggs

Table 9, Table 10 and Table 11 show the residual concentrations of TIM in the whole egg, yolk, and egg white at different points in time.

The TIM single administration group: The peak concentrations of TIM in the egg samples were 68.97 ± 19.08 μg/kg on the 3rd day of administration (Table 9). The peak concentration of TIM in samples was 72.59 ± 17.59 μg/kg on the 1st day after drug withdrawal. On the 3rd day after stopping the medication, the concentration of TIM decreased to 57.12 ± 13.06 μg/kg. The TIM content was 41.01 ± 10.51 μg/kg and 24.87 ± 8.26 μg/kg on the 8th and 10th days after drug withdrawal, respectively, and the TIM content was 12.95 ± 4.05 μg/kg on the 12th day (Table 10). The elimination of the TIM residue was completed on the 15th day, and the t_1/2_ was 2.15 d (Table 11).

The combined administration group: The peak concentrations of TIM in the egg samples were 75.36 ± 16.27 μg/kg on the 3rd day of administration (Table 9). The peak concentration of TIM in the samples was 78.96 ± 10.96 μg/kg on the 1st day after drug withdrawal. On the third day after stopping the medication, the concentration of TIM decreased to 66.02 ± 11.51 μg/kg. The TIM content was 30.65 ± 9.19 μg/kg and 11.69 ± 5.33 μg/kg on the 8th and 10th days after drug withdrawal, respectively (Table 10). The elimination of the TIM residue was completed on the 15th day, and the t_1/2_ was 1.46 d (Table 11). On the whole, the results showed that the distribution concentration of TIM in the combination group was significantly lower than the residual concentration of TIM in the single-drug group, and the elimination rate of TIM in the combination group was significantly faster than that in the single-drug group (Figure 4).

With time as the horizontal axis and the TIM content in the egg samples as the vertical axis, the fitting was carried out. The drug concentration at the next three time points where the drug could be detected was logarithmic and then the regression analysis was made with time to obtain the elimination parameters of TIM in the three samples. If the concentration was detected at less than three time points, the elimination parameters could not be calculated.

## 4. Discussion

Substances such as the protein and fat in egg samples greatly affect the determination of drugs. Sample pretreatment is crucial to resolving this issue. In order to reduce the interference of background compounds and reduce the damage and consumption of chromatographic instruments, the target analytes are usually extracted, purified, concentrated and filtered, and then detected. Currently, the simultaneous detection of EF, CIP, and TIM residues has few reports, but the research on a single-drug detection method has been quite mature. For the extraction of fluoroquinolones from eggs, a phosphate buffer solution, acetonitrile phosphate, trichloroacetic acid [30], and acetonitrile were used as extraction solutions and purified by the solid-phase extraction column [31]. This study compared the extraction effect of acidified acetonitrile, acetonitrile, trichloroacetic acid, and phosphate buffer on the basis of previous literature reports. The results found that the extraction effect of acetonitrile (containing 1% formic acid) was much higher than that of the other extraction reagents, and the reagent pH value had a substantial influence on the extraction rate of TIM. An acetonitrile (containing 1% formic acid) solution can assist in the extraction of TIM in egg samples by avoiding the interaction between TIM and other components, such as proteins and sugars. In this study, the purification effects and recovery rates of the liquid–liquid extraction, PRiME HLB, HLB, and C_18_ extraction were compared. The results of the study demonstrated that the HLB solid-phase extraction column was more suitable for the purification of the egg sample matrix, with a stronger purification effect, a higher recovery rate, and higher extraction efficiency, and the Prime HLB did not require balance activation. Finally, 80% acetonitrile of the same volume was selected for pre-washing and cleaning before loading so that the sample was completed through the column under the action of gravity only. Therefore, this study finally selected acetonitrile (containing 1% formic acid) as the extraction reagent and the HLB solid-phase extraction column to perform the role of purification.

The combination of EF and TIM has been reported to treat respiratory tract infections in poultry. Zhang et al. explored the interactions between EF and TIM on residual levels and drug use in broilers, separately and in combination. The results indicated that the metabolism of EF in broilers was affected by TIM by inhibiting CYP3A4 and increasing the residual concentration of EF. In addition, the time to eliminate EF has been extended [26]. However, the co-use of EF and TIM in laying hens is rarely reported. Therefore, we investigated the interaction and transformation of EF and TIM in laying hens individually and in combination.

Eggs are an important food for meeting the dietary needs of a growing population [32]. The therapeutic and prophylactic use of some veterinary medicinal products, such as antibiotics, can improve the efficiency of healthy poultry production, but irrational use can cause harm to animals and humans [33]. EF has a strong ester affinity, giving it a long half-life and slow metabolism after administration [34]. EF, when used in food animals, can be metabolized to ciprofloxacin and may remain in eggs, where the excessive ingestion of the residues can lead to the development of drug resistance and cause allergic reactions [35]. In addition, the irrational use of enrofloxacin in the breeding industry may lead to the risk of resistance transferring from animals to humans. Studies have shown that the use of enrofloxacin to treat disease in poultry may induce fluoroquinolone resistance in Campylobacter jejuni and can be transferred to humans, leading to treatment failure in humans [36]. In this study, according to the clinical recommendation of 5 days as the standard days of the EF single administration group and combined administration group, the EF soluble powder (0.75 g/L dose) was prepared into a solution for continuous drinking water administration, to study the rule of the EF residue elimination in egg samples after single and combined administration. The highest concentrations of EF and CIP were found on the 5th day of medication, with a C_max_ of 974.92 ± 441.71 μg/kg and 121.51 ± 14.38 μg/kg. The EF content was 121.54 ± 19.12 μg/kg and 68.64 ± 8.68 μg/kg at 8 d and 10 d after drug withdrawal, respectively. CIP was eliminated at 8 d and 15 d, and the half-lives of EF and CIP were 1.13 d and 1.39 d, respectively. This result is similar to that reported by Gbylik-Sikorska M et al. [12]. An analysis of the residues in the eggs of the combined administration group showed that the peak concentrations of EF and CIP in the egg samples were 1256.41 ± 226.10 μg/kg and 93.22 ± 22.79 μg/kg on the 5th day of administration. The EF content was 153.29 ± 25.48 μg/kg on day 8 and 89.51 ± 13.43 μg/kg on day 10 after withdrawal, and the EF concentration decreased to 58.71 ± 7.41 μg/kg on day 12. The elimination half-lives of EF and CIP were 1.17 d and 1.14 d. By analyzing the overall trend of EF in different groups, we found that the distribution concentration of the EF soluble powder in the single administration group was much lower than that in the combined administration group. The drug data showed that the residual concentration in the egg samples in the single administration group was lower than that in the combined administration group, and the metabolic consumption time was longer. The reason for this phenomenon may be that the metabolism of EF was inhibited after the combination of the EF soluble powder and TIM soluble powder; thus, the accumulation of EF in the eggs was increased, the elimination rate was slowed down, and the t_1/2_ was longer. However, the metabolite CIP produced decreases, the elimination rate increases, and the half-life becomes shorter.

As an animal-specific macrolide antibiotic, timicosin has a good effect on the treatment and prevention of pneumonia, pasteurellosis, and mastitis in the livestock and poultry industry due to its strong antibacterial activity and superior pharmacokinetics [37]. Its antibacterial activity is similar to that of tylosin, which can effectively inhibit Gram-positive bacteria, mycoplasma, spirochaete, and Gram-negative bacteria [38]. Timicosin has a good therapeutic effect on a respiratory tract infection caused by mycoplasma, etc. [39]. It has been approved for clinical use in Australia, Spain, the United States, France, and other countries [16]. Timicosin has fast absorption after internal administration or subcutaneous injection and a long maintenance time of effective blood concentration, with an especially high concentration in the lungs and milk [17]. In this study, 10% of the TIM soluble powder was prepared into a solution for continuous drinking water administration, which was set as the single administration group and combined administration group according to the clinical recommendation for 3 days, respectively, to study the rule of the residue elimination of TIM in the eggs. The residual concentration analysis of the single group showed that the peak concentration of TIM was 72.59 ± 17.59 μg/kg on the first day after withdrawal, and the C_max_ was 72.59 ± 17.59 μg/kg. The TIM content was 68.64 ± 8.68 μg/kg on the 10th day after withdrawal, and the TIM residue was eliminated on the 15th day, and the half-life of elimination was 2.15 days. The results in the single administration group of the TIM soluble powder were similar to those of Ji et al. [15]. The analysis of the residue in the eggs of the combined administration group showed that the concentration of TIM in the egg samples reached the peak at 78.96 ± 10.96 μg/kg on the first day after drug withdrawal, and the C_max_ was 78.96 ± 10.96 μg/kg. The TIM content was 24.87 ± 8.26 μg/kg on the 10th day after withdrawal, and the TIM residue was eliminated on the 15th day, and the half-life of the TIM elimination was 1.46 d. As a whole, the distribution concentration of TIM in the combined group was lower than the residual concentration in the single group, and the elimination rate of TIM in the combined group was faster than that in the single-dose group. We found that due to the co-use of the EF and TIM soluble powders, the metabolism of TIM was induced, resulting in a faster elimination rate and a shorter half-life of TIM in the eggs.

In conclusion, this study first optimized the sample pretreatment method, improved the extraction efficiency and recovery rate, and established the LC-MS/MS method that can simultaneously detect EF, CIP, and TIM. The residual amounts of EF and TIM in the egg samples were determined by the optimized method. The results confirmed that the combination of EF and TIM in laying hens resulted in a significant accumulation of EF in the eggs, a slower elimination rate, and a longer half-life than EF alone. Therefore, the co-use of TIM and EF in the poultry breeding industry must be reduced to ensure human and animal safety.

## Figures and Tables

**Figure 1 metabolites-13-00528-f001:**
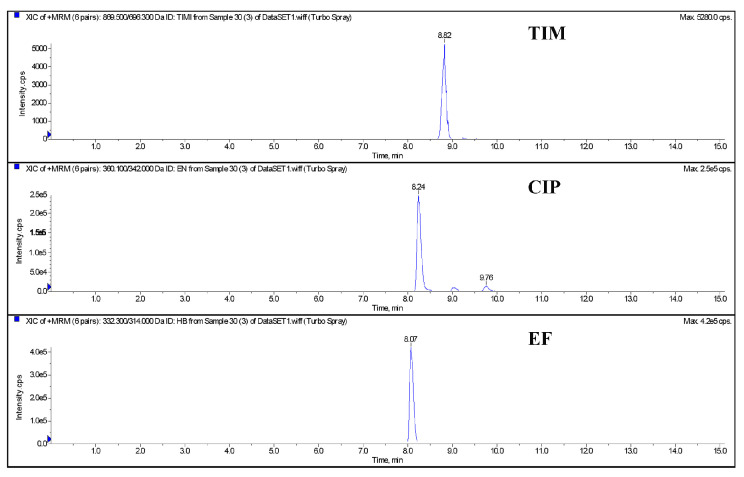
The chromatography of quantitative ions of control blank and spiked egg (TIM, CIP, and EF) by LC-MS/MS. Note: TIM (tilmicosin), EF (enrofloxacin), CIP (ciprofloxacin).

**Figure 2 metabolites-13-00528-f002:**
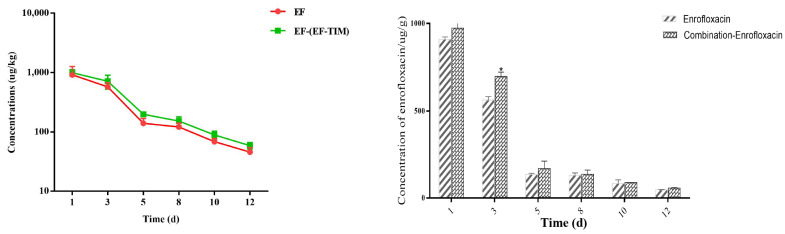
Time-lapse graphs of EF in eggs. Note: EF (enrofloxacin), TIM (tilmicosin). The data were expressed as mean ± SD. * *p* < 0.05 denoted statistical significance.

**Figure 3 metabolites-13-00528-f003:**
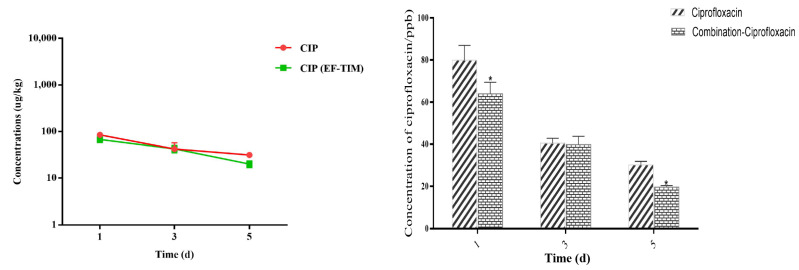
Time-lapse graphs of CIP in eggs. Note: CIP (ciprofloxacin), EF (enrofloxacin), TIM (tilmicosin). The data were expressed as mean ± SD. * *p* < 0.05 denoted statistical significance.

**Figure 4 metabolites-13-00528-f004:**
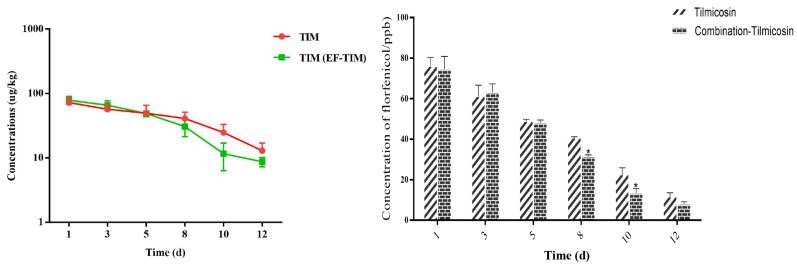
Time-lapse graphs of TIM in eggs. Note: EF (enrofloxacin), TIM (tilmicosin). The data were expressed as mean ± SD. * *p* < 0.05 denoted statistical significance.

**Table 1 metabolites-13-00528-t001:** Mass spectrometry parameters.

Compound	Parent Ion (m/z)	Daughter Ion (m/z)	Collision Pressure (EV)
TIM	869.4	174.2	62
		694.3	70
EF	360.1	315.97	19
		342	21
CIP	332.3	314	21
		231	35

Note: TIM (tilmicosin), EF (enrofloxacin), CIP (ciprofloxacin).

**Table 2 metabolites-13-00528-t002:** Standard curve equations and related coefficients.

Sample	Linear Equation	Coefficient of Correlation R	Range of Linearity (μg/L)
EF	Y = 14,808X + 134,620	0.9935	10~500
CIP	Y = 5296.7X + 129,506	0.9958	10~500
TIM	Y = 236.47X − 1.8887	0.9992	10~500

Note: TIM (tilmicosin), EF (enrofloxacin), CIP (ciprofloxacin).

**Table 3 metabolites-13-00528-t003:** The average recoveries and RSD for CIP, EF, and TIM in eggs (%).

Sample	Concentration of Addition (μg/kg)	Recovery Rate (%)					Average Recovery Rate in Batch (%)	Coefficient of Variation in Batch (%)	Average Inter-Lot Recovery Rate (%)	Coefficient of Variation between Batches (%)
		1	2	3	4	5				
CIP	10	85.25	73.34	83.25	69.02	87.77	79.72	9.10	81.28	2.71
		79.33	79.03	74.90	80.81	87.57	80.33	5.13		
		76.97	85.58	86.63	86.25	83.58	83.80	4.27		
	50	79.31	85.92	81.84	85.01	83.33	83.08	2.83	84.39	1.54
		86.67	78.51	80.52	90.63	92.08	85.68	6.27		
		93.36	72.40	77.06	89.57	89.61	84.40	9.65		
	100	80.07	86.11	79.65	79.22	91.19	83.25	5.65	87.06	3.88
		93.71	80.35	96.43	83.01	94.87	89.67	7.40		
		94.45	79.55	94.55	80.26	92.44	88.25	7.77		
EF	10	81.82	88.48	88.55	86.36	77.26	84.50	5.17	80.12	4.73
		70.31	80.26	85.20	78.09	76.57	78.08	6.22		
		79.25	78.46	78.27	75.52	77.42	77.78	1.63		
	50	87.59	92.87	90.48	93.33	90.41	90.94	2.26	89.41	2.39
		80.09	97.87	90.67	83.75	99.25	90.33	8.35		
		85.90	82.17	95.75	87.90	83.16	86.98	5.56		
	100	85.32	90.08	93.08	94.33	96.41	91.84	4.19	92.33	1.04
		87.68	94.96	90.79	85.44	99.68	91.71	5.57		
		94.67	97.33	82.36	88.44	104.38	93.43	8.07		
TIM	10	76.11	85.65	87.54	83.61	74.48	81.48	6.42	82.39	4.59
		69.65	80.90	89.56	75.96	79.68	79.15	8.23		
		79.47	87.46	98.48	80.95	86.38	86.55	7.75		
	50	80.44	84.96	83.67	80.55	77.58	81.44	3.20	87.62	6.18
		88.84	83.66	97.35	88.68	99.19	91.54	6.37		
		94.37	87.34	82.24	92.37	93.07	89.88	5.01		
	100	82.38	79.84	81.22	79.19	91.35	82.80	5.34	90.08	7.28
		83.85	99.49	89.58	86.97	99.77	91.93	7.12		
		86.89	102.38	94.66	93.65	99.99	95.51	5.65		

Note: CIP (ciprofloxacin), EF (enrofloxacin), TIM (tilmicosin).

**Table 4 metabolites-13-00528-t004:** Residual concentrations of EF during EF soluble powder in eggs (mean ± SD, *n* = 10) (µg/kg).

Time (d)/Sample Number (d)	EF Group	CIP Group
1 (*n* = 10)	179.87 ± 30.14	15.60 ± 2.84
2 (*n* = 10)	408.38 ± 136.13	36.83 ± 13.27
3 (*n* = 10)	804.41 ± 112.19	52.82 ± 8.06
4 (*n* = 10)	866.44 ± 14.85	80.33 ± 10.83
5 (*n* = 10)	974.92 ± 441.71	121.51 ± 14.38

Note: ND is less than the LOQ or not detected. CIP (ciprofloxacin), EF (enrofloxacin), TIM (tilmicosin).

**Table 5 metabolites-13-00528-t005:** Residual concentrations of EF after EF soluble powder in eggs (mean ± SD, *n* = 10) (µg/kg).

Time (d)/Sample Number (d)	EF Group	CIP Group
1 (*n* = 10)	919.73 ± 346.5	84.90 ± 11.26
3 (*n* = 10)	577.12 ± 101.85	42.12 ± 15.42
5 (*n* = 10)	140.20 ± 29.05	31.37 ± 4.93
8 (*n* = 10)	121.54 ± 19.12	ND
10 (*n* = 10)	68.64 ± 8.68	ND
12 (*n* = 10)	45.78 ± 8.65	ND
15 (*n* = 10)	ND	ND

Note: ND is less than the LOQ or not detected. CIP (ciprofloxacin), EF (enrofloxacin), TIM (tilmicosin).

**Table 6 metabolites-13-00528-t006:** Residual concentrations of EF during EF soluble powder combined with TIM soluble powder in eggs (mean ± SD, *n* = 10) (µg/kg).

Time (d)/Sample Number (d)	EF Group	CIP Group
1 (*n* = 10)	264.94 ± 43.74	16.61 ± 2.86
2 (*n* = 10)	425.62 ± 87.86	40.56 ± 12.74
3 (*n* = 10)	723.15 ± 99.88	54.40 ± 9.40
4 (*n* = 10)	955.39 ± 128.82	76.98 ± 14.08
5 (*n* = 10)	1256.41 ± 226.10	93.22 ± 22.79

Note: ND is less than the LOQ or not detected. CIP (ciprofloxacin), EF (enrofloxacin), TIM (tilmicosin).

**Table 7 metabolites-13-00528-t007:** Residual concentrations of EF after EF soluble powder combined with TIM soluble powder in eggs (mean ± SD, *n* = 10) (µg/kg).

Time (d)/Sample Number (d)	EF Group	CIP Group
1 (*n* = 10)	997.97 ± 122.85	67.80 ± 9.25
3 (*n* = 10)	715.55 ± 190.01	42.58 ± 7.43
5 (*n* = 10)	199.15 ± 17.05	20.10 ± 3.20
8 (*n* = 10)	153.29 ± 25.48	ND
10 (*n* = 10)	89.51 ± 13.43	ND
12 (*n* = 10)	58.71 ± 7.41	ND
15 (*n* = 10)	ND	ND

Note: ND is less than the LOQ or not detected. CIP (ciprofloxacin), EF (enrofloxacin), TIM (tilmicosin).

**Table 8 metabolites-13-00528-t008:** Elimination parameters of EF in eggs.

Groups	Drugs	Elimination Equation	K (d^−1^)	t_1/2_ (d)
Single group	EF	C = 1504.4 e^−0.615 t^	0.615	1.13
	CIP	C = 130.52 e^−0.498 t^	0.498	1.39
Combined administration	EF	C = 1740.1 e^−0.59 t^	0.59	1.17
	CIP	C = 130.57 e^−0.608 t^	0.608	1.14

Note: EF (enrofloxacin), CIP (ciprofloxacin); K: eliminate constants, t_1/2_: the half-life of elimination.

**Table 9 metabolites-13-00528-t009:** Residual concentrations of TIM in eggs (mean ± SD, *n* = 10) (µg/kg).

Time (d)/Sample Number (d)	TIM Group	Combined Administration Group
1 (*n* = 10)	21.83 ± 8.18	20.76 ± 3.72
2 (*n* = 10)	47.44 ± 17.53	33.34 ± 18.30
3 (*n* = 10)	68.97 ± 19.08	75.36 ± 16.27

Note: ND is less than the LOQ or not detected. CIP (ciprofloxacin), EF (enrofloxacin), TIM (tilmicosin).

**Table 10 metabolites-13-00528-t010:** Residual concentrations of TIM in eggs (mean ± SD, *n* = 10) (µg/kg).

Time (d)/Sample Number (d)	TIM Group	Combined Administration Group
1 (*n* = 10)	72.59 ± 17.59	78.96 ± 10.96
3 (*n* = 10)	57.12 ± 13.06	66.02 ± 11.51
5 (*n* = 10)	49.49 ± 16.52	49.05 ± 5.27
8 (*n* = 10)	41.01 ± 10.51	30.65 ± 9.19
10 (*n* = 10)	24.87 ± 8.26	11.69 ± 5.33
12 (*n* = 10)	12.95 ± 4.05	8.73 ± 1.42
15 (*n* = 10)	ND	ND

Note: ND is less than the LOQ or not detected. EF (enrofloxacin), TIM (tilmicosin).

**Table 11 metabolites-13-00528-t011:** Elimination parameters of TIM in eggs.

Groups	Drugs	Elimination Equation	K (d^−1^)	t_1/2_ (d)
Single group	TIM	C = 115.6 e^−0.323 t^	0.323	2.15
Combined administration	TIM	C = 161.47 e^−0.476 t^	0.476	1.46

Note: TIM (tilmicosin); K: eliminate constants, t_1/2_: the half-life of elimination.

## Data Availability

No new data were created or analyzed in this study. Data sharing is not applicable to this article.

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
