# Peer review of "Study on the Metabolic Transformation Rule of Enrofloxacin Combined with Tilmicosin in Laying Hens"

_metabolites, 2023, doi:10.3390/metabo13040528_

Round 1

Reviewer 1 Report

Comment to authors

Overall, the manuscript is well-written and informative. It provides a clear and concise background to the study, highlighting the importance of investigating the presence of EF and TIM residues in eggs and evaluating the potential risks to human health. The results are meaningful and clear and well-discussed.

Introduction:

In the first sentence, you may want to specify that China is one of the largest countries in the world in egg production, not just "one of the largest countries".

In the second sentence, you could consider rephrasing "per capita egg consumption rate in China is also gradually increased" to "the per capita consumption of eggs in China has been steadily increasing".

In the paragraph discussing EF, you could provide more context on why EF residues in eggs are particularly concerning. For example, what are the potential health effects of consuming eggs with EF residues, and how prevalent is this issue in the global food supply?

In the paragraph discussing TIM, you could provide more detail on why this antibiotic is important in animal husbandry, and how it is typically used.

In the final paragraph, you mention the need to investigate drug interactions between EF and TIM. However, it may be helpful to provide more background on why drug interactions are a concern, and how they can affect the safety and efficacy of antibiotics in animal and human health.

There are several statements in the introduction and discussion without references. For example in lines 51-53, Im in hesitate if this statement is correct. Please add a reference.

Line 74 this is incorrect “TIM is extremely sensitive to Mycoplasma gallisepticum”. In fact Mycoplasma gallisepticum is extremely sensitive to TIM treatment

Methods

The statistical analysis is unclear because the difference between the TIM and EF-TIM cant be evaluated based on linear regression or orthogonal contrast and just can cover the time effects. Please define how data analysis was performed.

Figure and table

All figures and tables should be self-standing. Please define and unify all the abbreviations in the footnote. Please use the same abbreviations for all tables and figures

Author Response

Dear reviewers,

Re: Manuscript ID:2256670 and Title:Study on the metabolic transformation rule of enrofloxacin combined with tilmicosin in laying hens.Thank you for your letter dated 16 February 2023.

We would like to thank  you for allowing us to submit a revised manuscript. We also thank the reviewers for the time and effort that they have put into reviewing the draft manuscript. Your professional and useful suggestions have enabled us to improve our paper. Based on the instructions provided in your letter, we uploaded the file of the revised manuscript.

Appended to this letter is our point-by-point response to the comments raised by the reviewers. The comments are reproduced and our responses are given directly afterward. If there are any questions about our work, please feel free to contact us. We hope that the revised manuscript is accepted for presentation at metabolites.

Thanks again for your valuable comments, which are very important and instructive for our future research, and hope that we can learn more from you!

Yours sincerely,

Xu Wang, Yanfei Tao

18 March 2023

Reviewer 1

Introduction:

  1. In the first sentence, you may want to specify that China is one of the largest countries in the world in egg production, not just "one of the largest countries".

Authors’ response: Thank you very much for your suggestions and comments. After reading this part carefully, "one of the largest countries" is amended as follows: "China is one of the largest egg producing countries". See page 1, line 37 for details.

  1. In the second sentence, you could consider rephrasing "per capita egg consumption rate in China is also gradually increased" to "the per capita consumption of eggs in China has been steadily increasing".

Authors’ response: Thank you very much for your suggestions and comments. We have made the following modifications to this part: "China's per capita egg consumption has been growing steadily". See page 1, line 39 for details.

  1. In the paragraph discussing EF, you could provide more context on why EF residues in eggs are particularly concerning. For example, what are the potential health effects of consuming eggs with EF residues, and how prevalent is this issue in the global food supply?

Authors’ response: Thank you very much for your suggestions and comments. We have modified this part in the manuscript. See page 14, line 466-477 for details.

  1. In the paragraph discussing TIM, you could provide more detail on why this antibiotic is important in animal husbandry, and how it is typically used.

Authors’ response: Thank you very much for your suggestions and comments. We have modified this part in the manuscript. See page 15, line 502-511 for details.

  1. In the final paragraph, you mention the need to investigate drug interactions between EF and TIM. However, it may be helpful to provide more background on why drug interactions are a concern, and how they can affect the safety and efficacy of antibiotics in animal and human health.

Authors’ response: Thank you very much for your suggestions and comments. In the introduction and discussion, we have covered in detail the background of drug interactions, existing problems and examples of EF and TIM combined application.  If the introduction of this part needs to be further supplemented, we will continue to modify it in the next revision. See page 2 line84-97, page 14, line 458-465for details.

  1. There are several statements in the introduction and discussion without references. For example in lines 51-53, Im in hesitate if this statement is correct. Please add a reference.

Authors’ response: Thank you very much for your suggestions and opinions. We have modified this part, see lines 53-55 for details.

  1. Line 74 this is incorrect “TIM is extremely sensitive to Mycoplasma gallisepticum”. In fact Mycoplasma gallisepticum is extremely sensitive to TIM treatment

Authors’ response: Thank you very much for your suggestions and comments. We have made the following modifications to this part: "Mycoplasma gallinarum is extremely sensitive to TIM, which has strong permeability in lung tissue, so TIM is one of the ideal drugs for clinical control of respiratory tract infections". See page 2, line 76-78 for details.

Methods

  1. The statistical analysis is unclear because the difference between the TIM and EF-TIM cant be evaluated based on linear regression or orthogonal contrast and just can cover the time effects. Please define how data analysis was performed.

Authors’ response: Thanks for your suggestion. We have checked the relevant literatures and made detailed answers according to our own experimental content. The differences between the TIM and EF-TIM were analysed by one-way ANOVA.  Statistical significance was set at *p < 0.05, and at **p < 0.01 was considered to be strongly significant. This part has been revised in the manuscript, See page 5, line 195-197 for details.

  1. All figures and tables should be self-standing. Please define and unify all the abbreviations in the footnote. Please use the same abbreviations for all tables and figures

Authors’ response:

Thank you very much for your suggestions and opinions. We have modified the abbreviations in the graphs and tables. For example page 7, line 258 .

Reviewer 2 Report

The authors present a manuscript titled Study on the metabolic transformation rule of enrofloxacin combined with tilmicosin in laying hens". In general, the work is well written and organized. The topic is certainly of interest to the readers and falls within the sector of the evaluation of the safety of use of antibiotic drugs, often divided in the scientific world. The experiments are clearly described and the results correctly reported.In order to improve the overall quality of the work, I suggest only the following changes:

In the abstract it could be of greater impact to insert the most significant numerical data, rather than the description of the same.

In the conclusions the authors could better highlight the advantages of the applied method with the possibility of using it also for other similar matrices.

Please check the English fluency along the text

Pg 13, Ln 357-358: This sentence was repeated twice. Please, correct

Author Response

Dear reviewers,

Re: Manuscript ID:2256670 and Title:Study on the metabolic transformation rule of enrofloxacin combined with tilmicosin in laying hens.Thank you for your letter dated 16 February 2023.

We would like to thank  you for allowing us to submit a revised manuscript. We also thank the reviewers for the time and effort that they have put into reviewing the draft manuscript. Your professional and useful suggestions have enabled us to improve our paper. Based on the instructions provided in your letter, we uploaded the file of the revised manuscript.

Appended to this letter is our point-by-point response to the comments raised by the reviewers. The comments are reproduced and our responses are given directly afterward. If there are any questions about our work, please feel free to contact us. We hope that the revised manuscript is accepted for presentation at metabolites.

Thanks again for your valuable comments, which are very important and instructive for our future research, and hope that we can learn more from you!

Yours sincerely,

Xu Wang, Yanfei Tao

18 March 2023

Reviewer 3

  1. In the abstract it could be of greater impact to insert the most significant numerical data, rather than the description of the same.

Authors’ response: Thank you very much for your suggestions and comments. We have modified the summary and added the data of significant significance. See the abstract of the manuscript for details.

  1. In the conclusions the authors could better highlight the advantages of the applied method with the possibility of using it also for other similar matrices.

Authors’ response: Thank you very much for your suggestions and opinions. We have made a supplementary explanation to the conclusion part of the method, which better reflects the superiority and reliability of the method. See the conclusion of the manuscript for details.

  1. Please check the English fluency along the text

Authors’ response: Thank you very much for your suggestions and comments.  The manuscript has been revised by native English speakers .

  1. Pg 13, Ln 357-358: This sentence was repeated twice. Please, correct

Authors’ response: Thank you very much for your suggestions and comments. We will delete the duplicated content in the manuscript.

Reviewer 3 Report

I read with interest the manuscript ID: metabolites-2256670 entitled “Study on the metabolic transformation rule of enrofloxacin combined with tilmicosin in laying hens” and appreciate the effort made by the authors.

The authors established a new liquid chromatoc-tandem mass spectrometry (LC-MS/MS) method to investigate the changes of Enrofloxacin (EF) and Tilmicosin (TIM) residues in eggs during the combination of EF and TIM, and to evaluate the feasibility of the combination of EF and TIM in poultry and whether it poses a potential threat to human health. They focused on an interesting topic, that is of great significance to analyze the use of drug combinations in eggs considering the increasing consumption of eggs worldwide. Although the experiment is appropriately designed and implemented, there are minor grammar, and syntax errors. Also, there were no ethical approvement or statements please proved it.

This study provides comprehensive and detailed information, and conclude that the use of EF and TIM in combination should be treated with greater care and supervision should be strengthened to avoid risks to human health.

Minor comments:

Line 256: Table 3 instead of table 3

Line 267: Tables 4-7 instead of Table 4-7

Line 270: Table 5 instead of table 5

Line 274: Table 8 instead of table 8

Line 278: Table 7 instead of table 7

Line 284: Figures 2 and 3 instead fig.2, fig.3

Line 306: Table 8 instead of table 8

Line 320: Tables 9-11 instead of Table 9-11

Lines 323, 331: Table 9 instead of table 9

Line 328: Table 10 instead of table 10

Line 336: Table 11 instead of table 11

Line 340: Fig.4 instead of fig.4

Lines 434, 435: Please revised the sentence!

Please insert the Conclusions section

Reference 14: Please verify the journal name and add the no. pages!

Author Response

Dear reviewers,

Re: Manuscript ID:2256670 and Title:Study on the metabolic transformation rule of enrofloxacin combined with tilmicosin in laying hens.Thank you for your letter dated 16 February 2023.

We would like to thank  you for allowing us to submit a revised manuscript. We also thank the reviewers for the time and effort that they have put into reviewing the draft manuscript. Your professional and useful suggestions have enabled us to improve our paper. Based on the instructions provided in your letter, we uploaded the file of the revised manuscript.

Appended to this letter is our point-by-point response to the comments raised by the reviewers. The comments are reproduced and our responses are given directly afterward. If there are any questions about our work, please feel free to contact us. We hope that the revised manuscript is accepted for presentation at metabolites.

Thanks again for your valuable comments, which are very important and instructive for our future research, and hope that we can learn more from you!

Yours sincerely,

Xu Wang, Yanfei Tao

18 March 2023

  1. Minor comments:

Line 256: Table 3 instead of table 3

Line 267: Tables 4-7 instead of Table 4-7

Line 270: Table 5 instead of table 5

Line 274: Table 8 instead of table 8

Line 278: Table 7 instead of table 7

Line 284: Figures 2 and 3 instead fig.2, fig.3

Line 306: Table 8 instead of table 8

Line 320: Tables 9-11 instead of Table 9-11

Lines 323, 331: Table 9 instead of table 9

Line 328: Table 10 instead of table 10

Line 336: Table 11 instead of table 11

Line 340: Fig.4 instead of fig.4

Authors’ response: Thank you very much for your suggestions and comments. We have modified this part of content, please see line 266, 280, 284, 288, 342, 369, 374, 375 for details.

2. Lines 434, 435: Please revised the sentence!

Authors’ response: Thank you very much for your suggestions and comments. After careful review of the article, we decided to delete this part of irrelevant content, so as to make the article more concise and smoother.

3. Please insert the Conclusions section

Authors’ response: Thank you very much for your suggestions and comments. Please refer to line 529 for the conclusion.

4. Reference 14: Please verify the journal name and add the no. pages!

Authors’ response: Thank you very much for your suggestions and comments. We have carefully checked the missing page numbers of the references in the manuscript and made supplements. Refer to Reference 14 at line 596.